

# Flood risk assessment through large-scale modeling under uncertainty

Luciano Pavesi[1], Elena Volpi[1], and Aldo Fiori[1]

[1]DICITA, Roma Tre University, Rome, Italy

**Correspondence:** Luciano Pavesi (luciano.pavesi@uniroma3.it)

**Abstract.** The complexity of flood risk models is intrinsically linked to a variety of sources of uncertainty (hydrology, hydraulics, exposed assets, vulnerability, coping capacity, etc.) that affect the accuracy and reliability of the analyses. Estimating the uncertainties associated with the different components allows us to be more confident in the risk values on the ground, thus providing a more reliable assessment for investors, insurance and flood risk management purposes. In this study, we investigate the flood risk of the entire Central Apennines District (CAD) in Central Italy using the laRgE SCale inUndation modEl - Flood Risk, RESCUE-FR, focusing on the interaction between the uncertainty of the hydraulic Manning parameter and the risk variability. We assess the coherence between the quantile flood risk maps generated by our model and the official risk maps provided by the CAD authority and focusing on three specific zones within the CAD region. Thus, RESCUE-FR is used to estimate the Expected Annual Damage (EAD) and the Expected Annual Population Affected (EAPA) across the CAD region and to conduct a comprehensive uncertainty analysis. The latter provides a range of confidence of risk estimation that is essential for identifying vulnerable areas and guiding effective mitigation strategies.

## 1 Introduction

Floods are one of the world's most devastating natural disasters, posing a significant threat to human life, infrastructure, economy and the environment (Doocy et al., 2013; Rentschler et al., 2022; Llasat et al., 2009). Defined as the temporary inundation of land not normally covered by water (Sayers et al., 2013), floods have shaped human civilization throughout history and continue to challenge societies worldwide. In recent years, the frequency and severity of flood events have increased due to factors such as climate change, land use change and population growth (Schilling et al., 2014; Blöschl et al., 2017; Cred, 2020).

Understanding flood risk, defined as the probability of a flood event combined with its potential adverse consequences, is paramount to effective disaster preparedness, response and mitigation strategies (EU Floods Directive, 2007). Flood risk assessment plays a crucial role in assessing the vulnerability of communities, infrastructure and ecosystems to flood events. It involves estimating the likelihood and potential impact of flooding (defined as hazard), taking into account the exposure (the presence of people, assets and systems in hazard zones) and the vulnerability (the susceptibility of these elements to flood damage) (UNISDR, 2009). Accurate flood risk assessment informs land use planning, emergency management and



25 infrastructure development, helping to identify high risk areas and prioritize mitigation measures (Falter et al., 2014; Convertino et al., 2019).

Despite its importance, flood risk assessment presents several challenges, particularly at large spatial scales that are fundamental for comparative analyses. Traditional methods for large scale analysis often rely on empirical or simplified models that may overlook important spatial variability and uncertainty in flood hazard and impact estimates (Alfieri et al., 2016; 30 Vorogushyn et al., 2018). In addition, the dynamic nature of flood risk, influenced by socio-economic changes and climate variability, adds further complexity to long-term assessments (Field, 2012).

Latest advances in hydrology and hydrodynamics modeling have enabled more accurate and detailed assessments of flood risk at larger spatial scales. These models provide improved representations of flood hazard, ranging from empirical methods to detailed hydrodynamic simulations (Manfreda et al., 2011; Schumann et al., 2013). Among these, LISFLOOD stands out as 35 a widely used model for large-scale flood inundation mapping. It provides valuable insights into flood extent and depth (Bates et al., 2010). In addition, the advances in large-scale flood modeling have contributed significantly to the assessment of flood risk at the continental scale. The Catchment-based Macro-scale Floodplain Model (CaMa-Flood), developed by Yamazaki et al. (2011), is an example of this progress, using advanced hydrodynamic simulations to capture the complex interactions between precipitation, land surface characteristics and river systems. High-resolution topographic data, coupled with advanced 40 modeling techniques, allow for more precise delineation of flood extent and depth, enhancing the reliability of flood risk assessments (Sampson et al., 2015; Yan et al., 2015).

Despite these improvements, flood hazard estimates are still quite uncertain. Uncertainty generally affects the reliability of risk estimates and arises from a variety of sources, including the climate projections, the model parameters and also the socioeconomic factors ruling the estimate of losses (Dankers et al., 2014; Ward et al., 2014).

45 The estimation of large scale flood losses is a complex issue due to the different interactions between systems that are complex such as: urban, suburban, agriculture, etc. In fact Merz et al. (2010b) and Meyer et al. (2013) highlight different types of damage. These include, for example: direct damage (damage caused by contact between water and structures), indirect damage (damage to customers/suppliers who cannot access the flooded area), damage due to business interruption (e.g. due to broken tools as a result of the flood), intangible damage (loss of life, epidemics, environmental damage, etc.) and risk mitigation 50 costs (installation of mitigation infrastructure, maintenance, etc.).

Due to the difficulty of representing all of these types of damage in models, modelers generally limit themselves to assessing direct damage (Merz et al., 2010b) because it can be associated with various hydraulic variables provided by flood hazard models, such as water depth, velocity, etc. These types of direct flood damage models can have different levels of complexity in terms of the variables considered and the spatial scale. They can be univariate, assuming that flood damage is influenced 55 by only one variable (usually water depth), and multivariate, assuming that flood damage is influenced by several variables (Gerl et al., 2016). In terms of spatial scale, they can be micro-scale (the damage is assessed at the building scale), meso-scale (the damage is assessed at the spatial aggregation) and large-scale (the damage is assessed at the municipal, regional, national scale) (Merz et al., 2010b).





Quantifying the overall uncertainty in both hazard and impact estimates is essential to understand the robustness of results in flood risk assessment and management (USACE, 1992; Peterman and Anderson, 1999; Downton and Pielke, 2005; Dankers and Feyen, 2009; Alfieri et al., 2015, 2016). Incorporating estimation uncertainty allows a probabilistic assessment of risk, providing a confidence interval for the results obtained rather than a deterministic risk value. This is useful for informed decision-making (Kreibich et al., 2017; Merz and Thieken, 2009; Merz et al., 2010a), in particular where we do not have data to validate our own damage estimates (Figueiredo et al., 2018; Molinari et al., 2020). More generally, this type of estimation supports the decision-making needs of different stakeholders who may have different attitudes to risk or different cost-benefit ratios for risk reduction measures (Merz and Thieken, 2009).

In this paper, we show how the variability of the Manning roughness parameter can affect the risk estimate. We consider here the Manning parameter as representative of most of the uncertainty characterizing the hydraulic modeling of flood risk. This is only one of the sources of uncertainty that need to be assessed in this context, such as the uncertainty in the estimates of the hydrological discharge, the vulnerability curve, the exposed asset values, and so on. Assessing uncertainty from other sources could provide additional insights and improve the overall confidence; but, achieving this requires a balance between the spatial resolution and the uncertainty assessment.

In this context, the use of a large-scale, simple and computationally inexpensive flood hazard model facilitates the parametric uncertainty analysis. To this aim we use the hydrologic-hydraulic modeling approach named RESCUE (laRgE SCale inUndation modEl), which was suited for the specific scope of the probabilistic assessment of flooded areas (Pavesi et al., 2022). Coupled with a damage model, this provides an estimate of the uncertainty of the flood risk. The RESCUE framework, combined with the damage model, is called RESCUE-FR (RESCUE - Flood Risk).

Although this article proposes a case study for Italy, where some damage studies have been carried out for the specific geographic region (see, e.g., Molinari et al., 2014; Amadio et al., 2019), we decided to adopt here the continental damage model of Huizinga (2007) and Huizinga et al. (2017) in order to be able to extend our analysis to a large transnational context in the future works. In addition, we estimate the population affected by floods; for this scope, we make the common simplifying assumption that the population is fully affected if located in the flooded area, regardless of the water depth (Ward et al., 2013; Alfieri et al., 2015).

The innovative aspect of this work is twofold:

– providing a more complete view of flood risk on a large scale assessing the associated uncertainties, given by the use of large scale flood models; this approach is achievable, not through complex models requiring intensive computational processing and calibration parameters, but through simpler large-scale models that illustrate the trade-off between quantile uncertainty and spatial resolution/detail;

– developing the RESCUE-FR framework, which integrates simple, globally accepted hazard, vulnerability and exposure assessment methodologies to ensure wider applicability of flood risk estimates across regional boundaries.



The paper is structured as follows: in Section 2 we explain the methods of RESCUE-FR framework, in Section 3 we present the study area, in Section 4 we present the main results and discussions, and in Section 5 we draw the main conclusions of our work.

## 2   Methods

### 2.1   Flood risk analysis

Hazard, vulnerability and exposure are the three fundamental modeling components of flood risk. In this work, we provide a probabilistic assessment of risk by accounting for the uncertainty due to the hydraulic modeling parameter, denoted here as $n$. To this aim we make some simplifying assumptions; we assume that the probability of flooding (i.e. the hazard) and the hydraulic characteristics of the inundation (e.g. water depth and velocity) are fully determined by the probability distribution function of the peak discharge, $p(Q)$, and of the Manning roughness parameter, $p(n)$ (where $p(.)$ indicates the probability density function), where $Q$ and $n$ are independent of each other. Moreover, we assume that the damage, $D$ (which results from the product of vulnerability and exposure), depends only on the water depth in the flooded area, $h$. Thus, risk $R$ can be described by the following equation

$$R = \int_\Omega R_n \, p(n)\mathrm{d}n = \int_\Omega \left( \int_{Q^*}^{\infty} D(h(Q,n))p(Q)\mathrm{d}Q \right) p(n)\mathrm{d}n \tag{1}$$

where $Q^*$ is the minimum peak discharge value producing damages and $R_n = \mathrm{E}_Q\left[D|n\right]$ (with $\mathrm{E}_X\left[.\right]$ denoting expectation relative to the probability distribution function of the variable $X$) is the expected annual economic loss in a given year, i.e. the Expected Annual Damage (EAD), and for a specific value of $n$ in the range of variability $\Omega$. Thus, $R = \mathrm{E}_n\left[R_n|n\right]$ is the average EAD (respect to the probability distribution of $n$) and, like $R_n$, is expressed in monetary terms.

Similarly, we can estimate the average population exposed to the effects of flooding in a given year, i.e. the Expected Annual Population Affected (EAPA), which is instead expressed in number of inhabitants. We denote $\Theta_n = \mathrm{E}_Q\left[\eta|n\right]$ the EAPA for a specific value of $n$ in the range of variability $\Omega$ and $\Theta = \mathrm{E}_n\left[\Theta_n|n\right]$ the average EAPA. These are obtained by substituting the damage function $D(h(Q,n))$ in Eq.(1) with the function $\eta(Q,n)$ counting the number of affected inhabitants as function of the hydrological load $Q$ and of the Manning parameter value $n$. As previously mentioned, we assume that the affected population does not depend on the water depth.

Note that Eq. (1) provides the average with respect to $n$ variability of EAD or EAPA in any point of the flooded area. Results can be integrated in space at the desired aggregation scale. Given the main objective of this work, in the application of the framework to the case study (sections 3 and 4) we particularly focus on the probability distribution of $R_n$ and $\Theta_n$ at different spatial scales.

Finally, we used a simplified approach to account for flood protection works. In large-scale flood hazard models, the resolution of the DTM is often insufficient to represent flood defense structures, and comprehensive databases detailing such defenses are often not available for manual integration. Previous studies have used various simplified approaches, such as assuming a





uniform level of protection (Rojas et al., 2013), or using the Gross Domestic product (GDP) per capita (Feyen et al., 2012) or the exposure (Zanardo and Salinas, 2022) as a proxy for level of protection. We adopt here a uniform level of protection by assuming the threshold value $Q^*$ in Eq. (1) equal to design return period of the flood protection works.

In the next subsections we describe in detail the methodology of the three component of the RESCUE-FR framework.

## 2.2   The hazard model: RESCUE

RESCUE is a large-scale inundation model that allows the assessment of flood hazard over large areas (Pavesi et al., 2022). It is a hybrid model combining geomorphological and hydrological-hydraulic approaches and consists of four distinct components.

1. Geomorphological analysis: using a Digital Terrain Model (DTM), the river network is extracted and segmented into
nodes and reaches. This segmentation helps to identify critical points such as breach nodes or channel heads. Each river segment is divided into equal lengths where possible.

2. Cross-section definition: average cross sections are defined for each reach identified in step 1. Key hydraulic properties such as wetted perimeter and area are derived using the Height Above Nearest Drainage (HAND) model (Nobre et al., 2011) as described in Zheng et al. (2018). These properties, together with water levels, are used to calculate key parame-
ters such as the mean wetted area and the hydraulic radius. Manning's equation is then used to generate the rating curve for each segment.

3. Hydrological load: the discharge in all river reaches for a given return period is calculated using the rational formula. This approach requires minimal information on the hydroclimatic conditions of the catchment and includes parameters such as the runoff coefficient and the critical rainfall intensity derived from an Intensity-Duration-Frequency (IDF) curve.

4. Flood map generation: flood maps are generated by solving the 1D hydrodynamic model under steady-state conditions. Starting from downstream boundary conditions, the gradually-varied-flow equation is solved along the river network to calculate water profiles for each segment. Water levels at segment nodes are interpolated and spatially propagated using the HAND map to identify flooded areas based on the terrain elevation and the water depth.

    The framework produces the extent of the flooded area for specific peak discharge values, corresponding to given return
period scenarios, and the related values of the water depth $h(Q, n)$ in any DTM cell of the area. To solve the integrals in Eq. (1), that is to account for the probability distribution of the peak discharge and the uncertainty associated with the Manning roughness parameter $n$, a Monte Carlo analysis is performed, resulting in several flood maps for each return period and Manning values.

## 2.3   Probabilistic mapping of flood hazard scenarios

The general approach to probabilistic flood mapping involves the evaluation of the statistical distributions for both model inputs (the peak discharge $Q$) and the parameters. Samples are taken simultaneously from each distribution and used as inputs to numerous model simulations. For each simulation, the inundated area is delineated and the flood inundation is defined





in probabilistic terms by analyzing the distribution of the results (e.g. the water depth). In RESCUE we incorporate only the statistical variability of the Manning roughness parameter $n$, considering it as representative of most of the uncertainty

characterizing the hydraulic modeling of flood risk.

To account for the variability of the Manning coefficient, we perform a Monte Carlo analysis using Latin Hypercube Sampling (LHS), as described in McKay et al. (2000). Compared to the case of Pavesi et al. (2022), where a uniform distribution of the Manning's parameter $n$ was assumed to assess the adaptation of flood hazard in flood-prone areas to official maps, we identified two key points:

1. the use of a uniform distribution for the roughness parameter, giving equal weight to all roughness values in an almost wide distribution, resulted in a certain overestimation of the water depth in the channel and consequently in the floodplains;

  2. the elevation of the channel cells obtained from the DTM is uncertain due to the presence of a water body; this affects the estimate of the channel bottom slope, which often results in null or very low values, thus introducing an additional
source of uncertainty in the estimation of the rating curve.

In general the discussion of the reliability of the rating curves and the key parameters influencing the rating curves is based on the model proposed by Zheng et al. (2018) and has been addressed in the literature by various authors (Garousi-Nejad et al., 2019; Godbout et al., 2019; Johnson et al., 2019; Ghanghas et al., 2022). In particular, from Ghanghas et al. (2022) results, changes in Manning roughness and channel slope predominantly affect rating curves during high flows, while the absence of

bathymetric data has a significant influence at low flows, with the effect decreasing as discharge increases. In RESCUE, our primary focus is on high flows, allowing us to neglect the bathymetric factor. In addition, the study by Johnson et al. (2019) suggests that wider ranges for varying the Manning parameter can be considered to improve the accuracy of rating curve estimate and the effectiveness of flood mapping models.

To address issues related to the channel bed slope parameter, particularly in low relief areas characterized by very low or
zero slopes, we decided to extend the lower limit of the Manning range parameter from 0.03 to $10^{-3}$ m$^{-1/3}$s, deviating from the approach used in Pavesi et al. (2022). In addition, to overcome the challenges associated with the uniform distribution of the Manning parameter, as discussed in point 1, we chose a normal probability distribution of the Manning parameter. For the probabilistic modeling of Manning, we assumed for $n$ a normal distribution with parameters $\mu = 0.0505$ m$^{-1/3}$s and $\sigma = 0.01921$ m$^{-1/3}$s. These parameters were chosen to ensure that 99% of the sample is positive and that the sample values
around the mean are consistent with literature values for Manning $n$ in floodplains as reported in Chow et al. (1988).

## 2.4   Flood damage model

The results from the hazard model, the water depth in each (DTM) cell of the flooded area $h(Q, n)$, can be combined with the damage model $D(h)$ to produce probabilistic estimates of risk, according to Eq. (1). The damage model follows the procedure described in Huizinga (2007), which is valid for all European countries. The choice of this type of damage model is guided by
the fact that the hazard model is implemented and solved at a large scale and therefore requires an consistent damage model





at the same scale (Merz et al., 2010b). In addition, the RESCUE model in combination with this damage model can also be used in an international context in a future work. This is introduced considering the product of the vulnerability, which strictly depends on the water depth, and the exposure, i.e. the exposed assets value.

Besides, with regard to estimating the affected population $\eta(Q,n)$, we consider the resident population affected for each positive water depth as in Alfieri et al. (2015) and Ward et al. (2013).

After calculating $D(h(Q,n))$ and $\eta(Q,n)$ maps, i.e. in all the DTM cells of the inundated areas generated by the hazard model, we can derive the $R_n$ and $\Theta_n$ maps by numerically solving the inner integral in Eq. (1) cell by cell; then, the empirical cumulative distribution functions (ECDFs) of $R_n$ and $\Theta_n$ are calculated for each cell.

In the following subsections the two components of the damage model and the population database used in the analyses are described in detail.

### 2.4.1 Vulnerability curves

The vulnerability assessment is based on relative (i.e. expressed in percentage terms) depth-damage functions at the European scale. As documented in Huizinga (2007) and Huizinga et al. (2017), these curves are derived from a comprehensive literature review of flood damage data and damage functions for 11 European countries. Five depth-damage curves are established for distinct economic sectors, namely residential, commercial, industry, infrastructure and agriculture. For those countries for which no information on flood vulnerability is available, the vulnerability curves are the mean value between the curves available for the European continent.

### 2.4.2 Exposure

The exposure is available for each of the 11 European countries and at the European scale; the latter is the average maximum assets damage between all the country in which we have information. For countries with no information on the overall value of the exposed assets, a methodology is applied to scale the average maximum losses (obtained from countries with available information) at national scale across different economic sectors using GDP (Gross Domestic Product) per capita PPS (Purchasing Power Standards) obtained from EUROSTAT. For further explanation see Huizinga (2007).

In this way, the maximum damage values are known for all countries. Since these values refer to the year 2007, it is necessary to take inflation into account and to update the maximum damage values for each country. This adjustment is made by calculating the ratio between the Consumer Price Index (CPI) of the current year and that of 2007. For further clarification the reader is referred to Huizinga et al. (2017).

This procedure is widely accepted as the standard approach for large-scale damage assessment in Europe, as shown by Rojas et al. (2013) and Merz et al. (2010b). The outlined procedure provides clear damage estimates for individual countries at the national level, but lacks differentiation at the sub-regional level (NUTS 3). The NUTS classification (Nomenclature of Territorial Units for Statistics) is a hierarchical system used to subdivide the economic territory of the EU for the purposes of the collection, development and harmonization of European regional statistics and the socio-economic analysis of regions, whereby NUTS 1 represents large socio-economic regions, NUTS 2 represents basic regions for the implementation of regional



policies and NUTS 3 represents small regions for specific diagnosis. Our study is particularly focused on this differentiation and therefore we have rescaled the maximum damage values to the NUTS 3 level. This rescaling was achieved by assigning weights derived from the ratio between the national GDP and the GDP of the respective province.

The assessment of exposed assets is carried out using the Corine Land Cover (CLC) map, which includes 44 different land cover classes. We adopt the methodology outlined in Huizinga (2007) to assess exposure on land cover grid data. This methodology is based on the integration of data from the European Land Use/Cover Area Frame Statistical Survey (LUCAS). Following the principles of Huizinga's methodology, a statistical mapping aligns the observed LUCAS with the CLC classes, resulting in a cross-tabulation. This cross-tabulation provides a comprehensive allocation of land use percentages within each CLC cell. Finally, we calculate a damage function for each land cover class as a linear combination of the corresponding damage functions for different land use classes. The weights in the sum are determined from Huizinga (2007).

### 2.4.3 Population

To assess the population at risk, we chose to use the European baseline dataset HANZE 2.0 (Paprotny and Mengel, 2023). HANZE (Historical Analysis of Natural Hazards in Europe) dataset was released in 2017, revised and expanded as HANZE 2.0 in 2023. It was the first comprehensive exposure dataset with resolution matching pan-European flood hazard maps, namely 100 m, covering the years 1870 to 2020, designed specifically to enable the analysis of exposure and land-use change within flood assessment studies. This decision was made in favor of the finer grid resolution compared to the 1 km resolution available in other European datasets.

## 3 Study Case: Central Apennine District

To illustrate and discuss the advantages of a probabilistic approach for large-scale flood risk mapping, we apply RESCUE-FR on the area pertaining to the Central Apennines River Basin District (CAD). The CAD is a geographical area of $42,298.22$ km$^2$ located in the heart of Italy (yellow area in Figure 1). It stretches from the Tyrrhenian Sea to the Adriatic Sea, including the Apennines, and comprises seven regions: Emilia-Romagna, Tuscany, Umbria, Latium, Marche, Abruzzi and Molise. As shown in Figure 1, Rome, Perugia, Pesaro, Ancona, Ascoli Piceno, Teramo, Pescara, Chieti, L'Aquila, Viterbo and Latina are among the most important cities (the boundary of each municipality is reported in the figure) in the district in terms of population and economic activity.

The CAD climate is Mediterranean, with rainfall concentrated in the autumn and winter months. The morphology is characterized by mountains, hills and valleys, shaped by the erosive action of the rivers and streams that cross the area. The most vulnerable areas are those located in the floodplains of rivers and their tributaries.

The CAD Authority provides the official flood risk map for the whole district through the PGRAAC (*Piano di Gestione del Rischio Alluvioni dell'Appennino Centrale*, i.e. the flood risk management plan of the Central Apennines). These maps are qualitative risk maps that can be considered as a reference for the assessment of the RESCUE-FR model. The PGRAAC maps are obtained by superimposing flood hazard maps (which result generally from different hydrologic methods and detailed

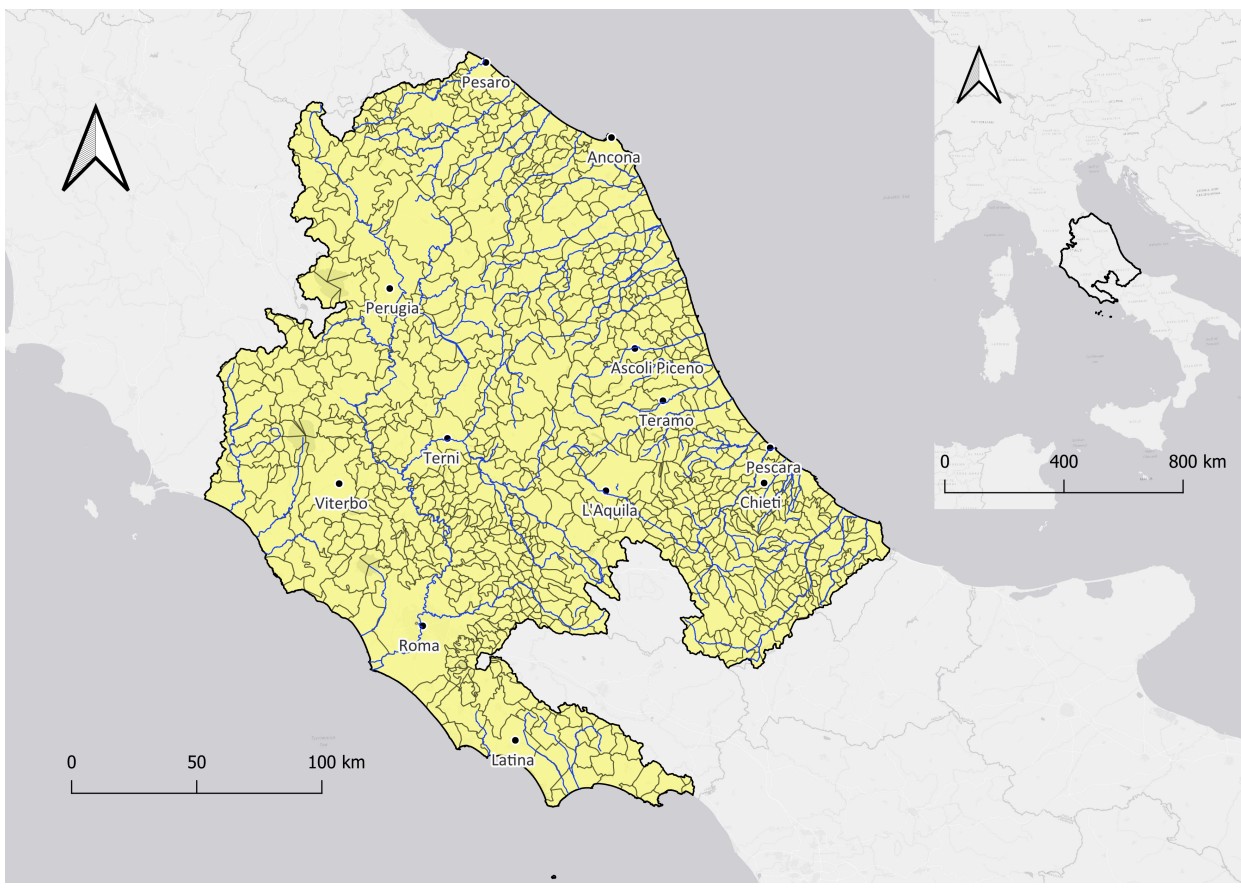

**Figure 1.** Area of interest: the Central Apennines District (Italy) divided in municipalities (black boundary); the main cities in terms of population and economic activity are reported (black dots).

hydrodynamic models) and a detailed land use vector map. These risk maps do not represent risk in economic terms, but are based on the identification of four qualitative risk classes: R1 (low risk), R2 (moderate risk), R3 (high risk), R4 (very high risk). The CAD Authority maps are based on a risk matrix that considers different return period flood maps and evaluates the land use within the flooded area. Regions with lower return periods and urban land use, with potentially high damage to buildings and people, have higher risk (R4 or R3). Conversely, areas with higher return periods and agricultural or forest land use have moderate (R2) to low risk (R1). The four classes, from R1 to R4, are in order of increasing risk severity.

## 3.1 Model setup

Identifying the exact location of rivers is crucial to flood risk assessment. Relying solely on an unconditioned DTM for river extraction can lead to inaccuracies, particularly in planimetry, which may lead to incorrect overlapping with land use types such as residential or others that are not appropriate. To address this issue, we used the MERIT Hydro dataset (Yamazaki et al.,





2019), which is hydrologically conditioned globally, ensuring a more accurate representation of river networks. This choice increases the reliability of the analysis by minimizing inaccuracies in river delineation and subsequent overlaps with land use categories for damage estimation.

The RESCUE model setup mirrors the configuration used in Pavesi et al. (2022) in terms of geomorphological analysis and hydrological discharge estimation. A threshold area for channelization of $A_c = 10$ km$^2$ and an average reach length of $\lambda = 1$ km are used for geomorphological analysis. The rational method uses the IDF curves at the regional scale based on the Two Components Extreme Value (TCEV) distribution provided by the VAPI project (Rossi and Villani, 1994). The hydrologic model also includes the modified Soil Conservation Service-Curve Number (SCS-CN) approach by Hawkins et al. (2010).

Only in the Adriatic coastal areas, where the SCS-CN method does not accurately reproduce the peak discharge, the rational approach is discarded. Therefore, to ensure reliable discharge estimation in these catchments, we integrated official basin-scale studies obtained from hydrologic-hydraulic reports available online (AUBAC, 2024).

We also deviate from Pavesi et al. (2022) for the hydraulic Manning distribution. We choose a normal distribution for the Manning parameter $n$ with a mean of $\mu = 0.0505$ m$^{-1/3}$s and a standard deviation of $\sigma = 0.01921$ m$^{-1/3}$s. These parameter choices were made on the basis of the discussions outlined in Section 2.3.

The analysis was carried out on the 25 main catchments defined by the Italian National Geoportal. In order to assess risk and damage scenarios in each catchment, we have discretized the peak discharge return period in chosen intervals ranging from 100 to 1000 year return period. In this regard, for each return period, we conducted 11 simulations using Latin Hypercube Sampling (LHS) to ensure that each sample was representative of a certain probability of occurrence of the Manning parameter.

To account for flood protection works, we assumed a uniform level of protection in all areas. In Italy the areas exposed to higher potential damage are generally protected by flood defenses up to a return period value of 200 year. Assuming a uniform flood protection return period of 200 year in the whole area may lead to a significant underestimation of the risk in those areas, whose extent is not negligible, that are effectively unprotected. For this reason, we look for a trade-off between these two issues and choose a uniform flood protection return period of 100 year. The choice of this return period is in agreement with the studies of Ward et al. (2013) and Rojas et al. (2013). Hence, we assume $Q^* = Q_{100}$ in Eq. (1) to mitigate the overestimation of flood risk associated with the lower return periods, where $Q_{100}$ is the $1 - 1/100 = 0.99$ quantile of the probability distribution of the peak discharge.

As described in section 2.4.1, we adopted the European scale vulnerability curves following the procedure outlined by Huizinga (2007). The NUTS 3 level was considered the most appropriate scale for our case study, which focused on the Central Apennines District. It was therefore essential to assess differences between different provinces rather than regions (NUTS 2 level). To ensure consistency with the methodology presented in Huizinga (2007), where a cross-tabulation between LUCAS and CLC data is presented, we have updated the maximum damage values for each category from Huizinga (2007) to current values adjusted for inflation. In addition, we have chosen to use the CLC 2000 dataset (Büttner et al., 2002) and refer to the land use percentages in each cell as reported in Huizinga (2007) report.



Finally, we refer to the baseline dataset HANZE 2.0 (Paprotny and Mengel, 2023) at 100 m spatial resolution to assess the
population at risk. This approach is consistent with established European methodologies, while ensuring an accurate representation of vulnerability curves tailored to our study area.

## 4    Results and discussion

In this Section, we present and discuss the results of the analysis of flood risk estimation uncertainty carried out on the CAD case study. First, we focus on evaluating the consistency between the flood risk maps generated by RESCUE-FR and the
official risk maps provided by the CAD authority. This comparison is essential to assess the reliability of RESCUE-FR; results are illustrated in Figure 2 for the whole CAD region.

In Figure 2, we compare the PGRAAC maps (panels a, c, e) with the RESCUE-FR risk maps at the 50-th quantile of risk (the median of $R_n$ distribution) (panels b, d, f), expressed in euro/year. Note that agricultural land use has significantly different maximum damage values compared to the other land use types, varying by about two orders of magnitude. To illustrate these
differences, we have chosen a logarithmic scale for risk visualization (per order of magnitude), in line with the qualitative approach used by the CAD authority. To simplify the representation and favor the comparison with PGRAAC maps, we assign three different risk classes based on the loss entity: C1 class risk value up to $10^2$ euro/year, C2 up to $10^3$ euro/year, above $10^4$ euro/year. Our classes, no matter how they are defined, do not correspond to the R1-R4 risk classed used by CAD that are not defined in numeric values.

Figure 2a-b focus on an area of the Upper Tiber River at the confluence with the Chiascio River near Perugia. Here, the two flooded areas (RESCUE and PGRAAC) differ significantly, especially along the Chiascio River. In particular, RESCUE tends to underestimate the flood extent for higher return periods compared to PGRAAC; this is due to errors in the DTM representation of the valley (results not shown). However, in the case of forest/agricultural land use, despite a significant underestimation of the flooded area, the error in risk assessment due to underestimation of the flooded area by the hazard model is not very
significant because the flooded area underestimated by RESCUE is in the low risk class R1 in PGRAAC risk map. This would be different if the land use was of a different type (residential, industrial, etc.), which would result in a higher risk class and make the local underestimation more relevant.

Figure 2c-d show a zoomed-in section of the middle Tiber Valley, revealing a very good agreement between the two maps in the transition from moderate/low-risk to high/very high-risk zones. However, we do notice some differences in high-risk
areas, which could be underestimated, particularly in zones adjacent to the river channel. This discrepancy is primarily due to the coarser resolution of the RESCUE-FR model. The CLC dataset, with a spatial resolution of 100 m, cannot capture detailed land use variations within a single cell compared to the land use vector map used by CAD Authority. Therefore, if a cell has a small percentage of residential or industrial land use but the majority is agricultural, it will be classified as agricultural damage. However, the resulting underestimation of risk compared to the total risk is almost negligible. Finally, in Fig. 2 e-f we show
almost the entire Musone river risk maps, where we can draw similar conclusions to the panel c-d.

**Figure 2.** Comparison between the PGRAAC maps (a, c, e) and the RESCUE-FR 50-th quantile of $R_n$ risk maps (b, d, f); panels a) and b) show a zoom on the upper Tiber River near the city of Perugia, panels c) and d) show a zoom on the middle Tiber valley near Rome, panels e) and f) show a zoom on the entire Musone River in the Marche Region. See the text the definition of the PGRAAC and RESCUE-FR risk classes.





In general, RESCUE-FR shows a consistent agreement with the official CAD maps, with the exception of areas that cannot be accurately mapped by the hazard model due to limitations of the DTM and the spatial resolution of the CLC, which may miss localized areas of land use. In the following we analyze and discuss the results of the uncertainty analysis.

Figure 3a shows an illustrative median $R_n$ risk map of a segment within the Aterno-Pescara basin, derived from the
RESCUE-FR model analysis. Similar to Figure 2, the legend is log-scaled to highlight the variation in risk between different land use types within cells. The purpose of this figure is to highlight the variability due to parameter uncertainty that is reflected in the $R_n$ probability curve, and to understand the added value of the uncertainty assessment. In general, three patterns of risk distribution were identified throughout the analysis. These are shown in Figure 3b, c and d. To illustrate these patterns, we select three specific cells with different land uses, different empirical cumulative distribution functions (ECDFs)
and different distances from the channel along a cross-section of the risk area.

In Figure 3b, we show the $R_n$ distribution of cell 1, which is a typical cell close to the main river channel and therefore always wet, with water depths often above 6 m (maximum damage). The distribution trend is asymptotic and there is no variation in the inter-quantile range. The risk values are the values of a cell which remains in a commercial/residential land use, such as CLC class 1.1.2. For a description of the Corine land Cover (CLC) classes see, e.g., Büttner et al. (2002).

In the panel c, we see a different pattern of the risk $R_n$ distribution, of a typical cell far from the main river channel that start to be wet from about the 7-th quantile and with a large variation in its inter-quantile range. The $R_n$ values are the values of a cell which remains in a agricultural land use such as CLC class 2.4.1. For this type of land cover class the risk value is very low compared to the land cover class representing the residential/commercial risk. The variability for agricultural land cover is therefore negligible in absolute economic terms.

Panel d, on the other hand, shows a typical cell which, despite being very close to the channel, is topographically much higher than the channel bottom and therefore only flooded when the water depth reaches very high levels. Here we can see that if we did not use such an approach to quantify the parametric uncertainty, we would greatly underestimate the risk associated with a certain area. In fact, in this case, the land use is commercial/residential and therefore the relative risk is very high if the cell is flooded. Without this approach, we would probably have rated the risk in this cell as zero.

To summarize, there is a considerable variability in the risk values provided by large-scale maps due to the uncertainty in the Manning parameter, embedding the parametric uncertainty of the hydraulic modeling approach. Thus, it is important to understand how these risk values, $R_n$, can be used effectively. A clear and comprehensive view of the extent of risk variability across the whole area is essential in order to identify the critical points across the area of interest, and thus delineate the areas where risk reduction efforts should be focused and their priority.

Therefore, in Figures 4 and 5 we present the main characteristics of the probability distribution of the expected annual damage $R_n$ and the expected annual population affected $\Theta_n$ aggregated at the municipal level; for both, we show the median and the 90-th quantile. Note that the boundary of each municipality is depicted in black as in Figure 1. It is important to stress that, as mentioned in Section 3.1, the analysis was carried out only for the 25 largest catchments. Since within some municipalities there are areas pertaining to catchments that are not simulated with RESCUE-FR, we have excluded these
municipalities from the discussion if the non-simulated area is 100% (gray municipalities in the figures). In all the other cases,


**Figure 3.** $R_n$ Empirical Cumulative Distribution Function (ECDF): a) zoom of the RESCUE-FR median risk map on the Aterno-Pescara River; b) local distribution in a cell near the main channel (commercial/residential land cover); c) local distribution in a cell far from the channel (agricultural land cover); d) local distribution of the risk of a cell that is very unlikely to get wet (commercial/residential land cover). For a description of the Corine land Cover (CLC) classes see, e.g., Büttner et al. (2002).

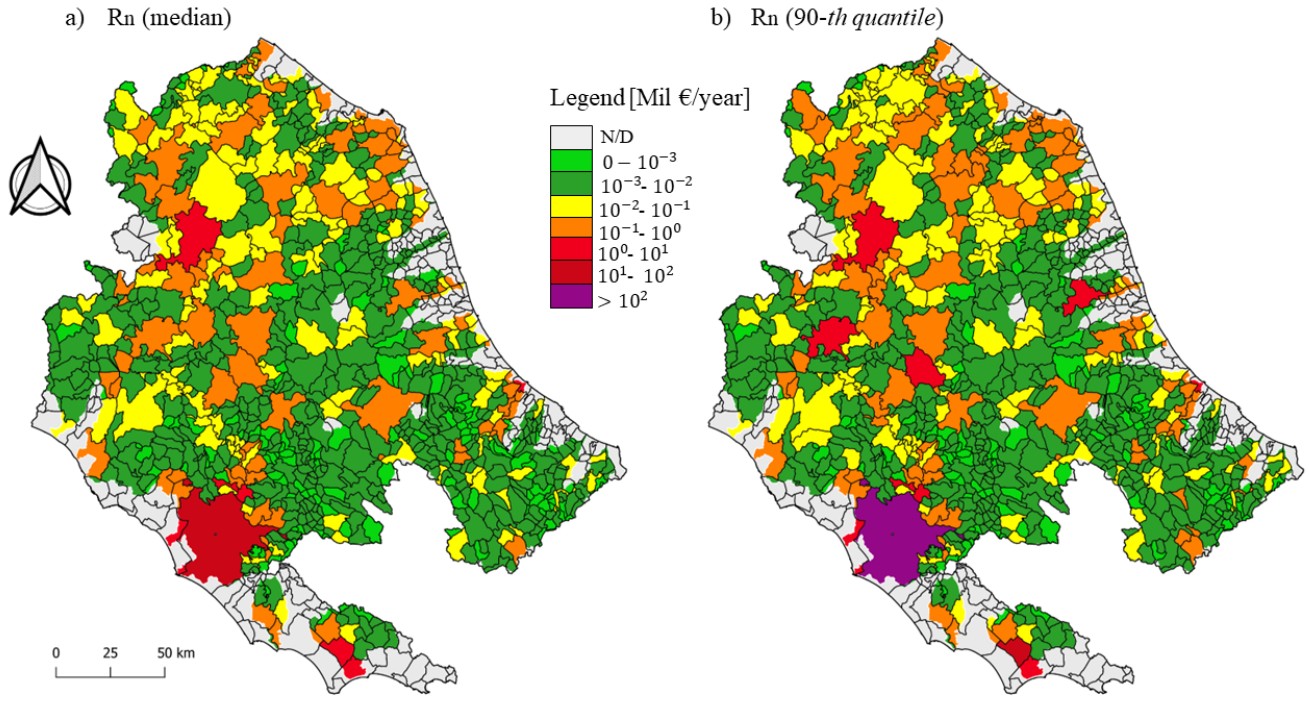

**Figure 4.** Expected annual damage $R_n$ distribution at the municipality scale: a) median quantile, b) 90-th quantile.

where only a fraction of the municipality area is not simulated, the risk (in terms of EAD or EAPA) in that municipality is generally underestimated. This is visible in the maps because the the non-simulated area within the municipality boundary is in gray color. The gray areas are concentrated in the coastal areas, where the catchments are generally small.

From the analysis conducted, it emerges that the median EAD in the CAD is approximately 137 M euro, with an average

exposed population of around 5,350 individuals annually. When quantile 90-th of the total EAD and EAPA distributions is assessed, these values increase by $34\%$ and $27\%$, respectively.

The values in Figures 4 and 5 show a wide range of annual losses in terms of EAD and EAPA. As expected, the highest risk values are found in the major economic and populated cities (as reported in Section 3), where there are more developed economic activities and larger resident population. Particularly Figure 4a shows that $63\%$ of the CAD exhibits a median risk

of less than 1,000 euro/year or even null. For another $27\%$, the risk appears to be less than 100,000 euro/year, while for the remaining $10\%$, it exceeds 100,000 euro/year. The reason for this large percentage at low risk is that the highest risks are in cities passing through main rivers, which have larger discharges than tributaries.

Moreover, by comparing figures 4a and 4b, it emerges that a different number of areas have a significant increase in EAD at quantile 90-th compared to the median. This allows to quickly identify areas of high uncertainty and carefully decide where to

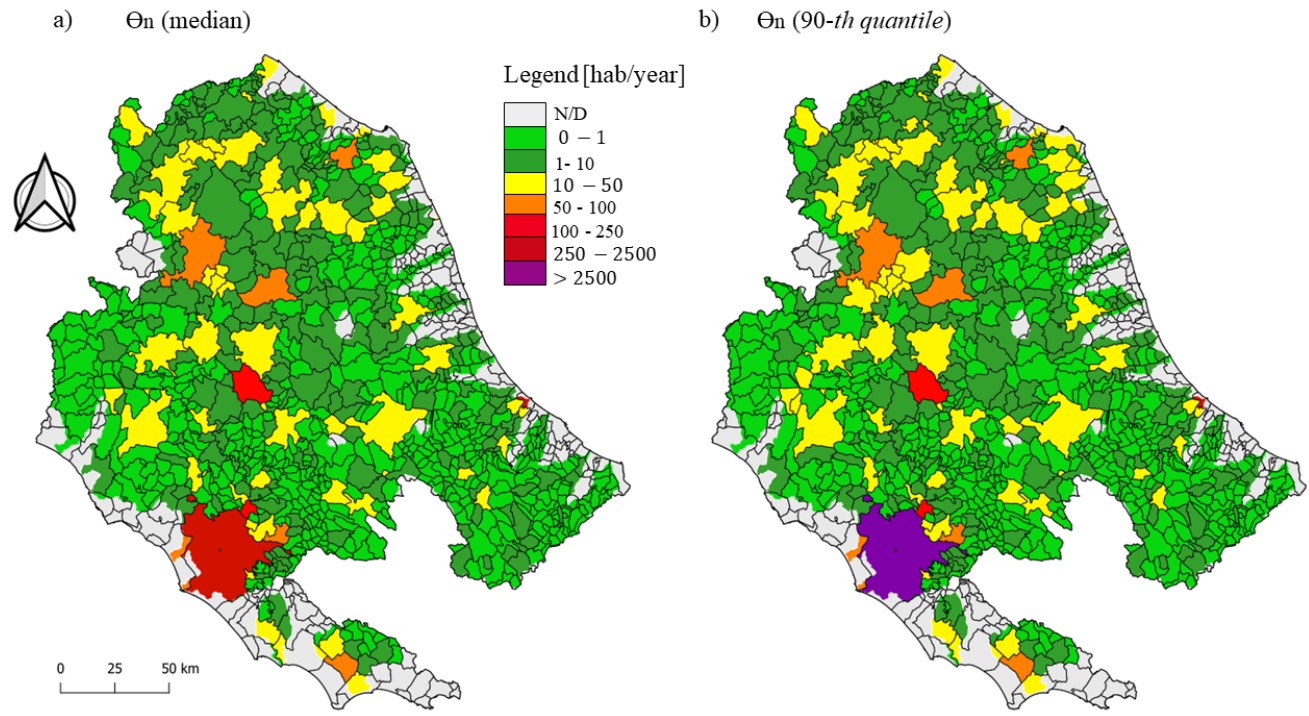

**Figure 5.** Expected annual population affected $\Theta_n$ distribution at the municipality scale: a) median quantile, b) 90-th quantile.

focus for further assessment or for risk management actions. This provides an immediate indication of areas that may require priority actions to reduce the impact of potential flood-related disasters.

As expected being the largest city in the area, the Municipality of Rome stands out as having the highest median risk, estimated at around 82 M euro/year. This corresponds to $60\%$ of the total risk in the CAD. This value is in line with the results of a previous study (Fiori et al., 2023), which provided an estimated risk of $24.3$ M euro/year, the same order of magnitude. The difference is justified considering that the risk estimate of Fiori et al. (2023) results from the implementation of a detailed two-dimensional hydrodynamic model, limited to the historic center of the city of Rome rather than the entire municipality. For the same area of the study mentioned above, our model has an average value of around 43 M euro. Despite a simplified representation of the phenomenon, the model is able to provide reliable estimates, also enabling the assessment of parametric uncertainty.

As for the population at risk, Figure 5a shows that $93\%$ of the CAD have an average population affected of less than 10 persons per year or no risk at all. For another $6\%$, the risk appears to be less than 50 persons exposed per year, while for the remaining $1\%$ it exceeds 2,500 persons exposed per year. Furthermore, it is interesting to note that in almost the $50\%$ of the simulated area the population at risk is equal to zero (light green areas). This can be explained by the fact that in the catchments that do not cross large urban centers, such as cities shown in Figure 1, the population is concentrated in the mountainous areas,





at high altitudes above the rivers and not affected by floods. Finally, the 90-th quantile does not show a significant variability in the population exposed at the municipal level, since this varies only in relation to the flooded area and the intersection of this area with the population distribution, without taking into account the variation in water depth.

## 5 Conclusions

In this study, we present and discuss the assessment of the parametric uncertainty in flood risk estimation performed with the large-scale risk model named RESCUE-FR. The model couples hazard assessment with economic damage estimation through widely recognized methodologies and can be easily extended to the entire European continent. More importantly, thanks to its simplicity, RESCUE-FR allows to investigate uncertainty without excessive computational efforts. We analyze here the effect of the variability of the Manning parameter, being representative of most of the different sources of uncertainty related to hydraulic modeling. Further, we demonstrate how the resulting distribution of risk can be used effectively for risk assessment and management.

The risk analysis was carried out for the CAD region (Central Italy). From this analysis we draw the following main conclusions.

– The RESCUE-FR model shows a consistent agreement with the qualitative risk maps provided by the CAD Authority. There exist areas of discrepancies, where we generally observe an underestimation of the risk, that is mainly due to the accuracy of the DTM and the limitations in the detailed mapping of land use changes given by the coarse resolution of the Corine Land Cover.

– Through the uncertainty analysis, the RESCUE-FR model provides a range of estimated flood risk (the expected annual damage), revealing significant variability in risk values depending on the land use considered.

– We identified three different patterns of risk distribution at the cell scale, depending on the distance from the canal, the orography and the different land use types. These patterns are useful to identify the critical points across the area of interest, where a single-value estimate of risk could significantly underestimate risk.

– In terms of affected population, the variability is lower as it depends mainly on the extent of flooding, which is less influenced by the Manning parameter with respect to water depth. In this case, the median value could be sufficient for the assessment of the expected annual population affected. Conversely, the expected annual damage has a higher variability being directly linked to water depth, requiring a thorough examination of the distribution quantiles for a comprehensive risk assessment of the whole area.

– Aggregating risk values at the municipal level provides a comprehensive overview of flood risk across the CAD region. This approach identifies areas of vulnerability and variability in risk levels, helping to prioritize risk reduction efforts.





    – The comparison with a previous study for the Municipality of Rome demonstrates the reliability of the RESCUE-FR

420       model in providing valuable information on flood risk, despite its simplified representation compared to more detailed

      hydrodynamic models.

We recall that the accuracy of RESCUE-FR estimates might be limited by the simplifying assumptions that, on the other hand, allow for an easy implementation of the uncertainty analysis. These include the spatial resolution of the maps used in input (DTM and CLC). Although these maps allow a consistent use of information at a larger scale, the accuracy of the maps

could be improved by a finer resolution at the cost of an increased computational load. Furthermore, the use of open databases containing the distribution of major defense structures, such as levees and dams, could improve the accuracy of risk estimate.

Finally, by assessing the uncertainty associated with individual parameters, we gain valuable insight into interpreting flood risk and understanding map results over large areas. However, it's important to recognize that uncertainty comes from multiple sources beyond hydraulic parameters. Therefore, extending our analysis to assess different sources of uncertainty, including

those related to hydrologic and socio-economic modeling, will make the analysis more complete, as it would take into account and compare different phenomena. This is far beyond the scope of this work, and will be object of future analyses.

*Data availability.* The hydrologically conditioned MERIT Hydro DTM is openly available at https://hydro.iis.u-tokyo.ac.jp/~yamadai/ MERIT_Hydro/. For damage estimation we used the Corine Land Cover (CLC) map from https://land.copernicus.eu/en/products/corine-land-cover/ clc-2000, the vulnerability curves for Europe available at https://publications.jrc.ec.europa.eu/repository/handle/JRC105688 and the CPI at

https://data.worldbank.org/indicator/FP.CPI.TOTL?locations=IT. The NUTS3 regions boundary is available at https://ec.europa.eu/eurostat/ web/nuts and the corresponding NUT3 regions GDP at current market prices at https://ec.europa.eu/eurostat/databrowser/view/nama_10r_ 3gdp/default/table?lang=en. The municipalities boundary are currently accessible at https://www.istat.it/it/archivio/222527, while the population dataset HANZE 2.0 is openly available at https://zenodo.org/records/6783023. Finally, the PGRAAC data that support the findings of this study are available at http://www.autoritadistrettoac.it.

*Author contributions.* L.Pavesi: Methodology, Formal Analysis, Data curation, Software, Visualization, Writing- Original Draft; E.Volpi: Conceptualization, Methodology, Writing- Reviewing and Editing. A.Fiori: Conceptualization, Methodology, Writing- Reviewing and Editing

*Competing interests.* The authors declare that no competing interests are present.



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
