# Peer review of "Flood risk assessment through large-scale modeling under uncertainty"

_Natural Hazards and Earth System Sciences, 2024_

## Author Response (AR1)

**"Flood risk assessment through large-scale modeling under uncertainty"**
by Luciano Pavesi, Elena Volpi, and Aldo Fiori

We wish to thank the Associate Editor and the two anonymous Reviewers who reviewed our paper. For ease of reference, we reproduce in the following the Reviewer's comments and our response in *italic characters*, as already published during the discussion.

Together with the revised version of the manuscript, we also send the Marked Manuscript in PDF in which all the changes in the text are marked (deleted in  characters, while new text is in blue characters). Line or page numbers refer to the Marked Manuscript; those highlighted in yellow indicate the line numbers where changes are incorporated into the manuscript.

**Reviewer #1**

The study presents a new approach to assessing flood risk, based on monetary assessments of damage and exposed population, considering a more simplified modeling, which allows its application on a large scale. The main novelty of the study is accounting the uncertainties associated with the Manning roughness parameter (n). The approach is innovative and useful to assist in the decision-making process to increase the resilience of localities to floods. The results presented interesting inferences and discussions. I really enjoyed reading the text and learning about the methodology applied. Below, I present some considerations that can be incorporated into the text to deepen the discussion and some doubts that I hope will be better clarified.

*Dear Reviewer,*

*Thank you for your careful and insightful review of our manuscript. We greatly appreciate the time and effort you have taken to provide such detailed comments and suggestions.*

The authors mention in the introduction some statements that can be better detailed to bring greater value. For example: "Traditional methods for large-scale analysis often rely on empirical or simplified models that may overlook important spatial variability and uncertainty in flood hazard and impact estimates" – such as?; "…and multivariate, assuming that flood damage is influenced by several variables" - which ones?

*Thank you for your comment, the text has been modified as follows:*

1. *Despite its importance, flood risk assessment presents several challenges, particularly at large spatial scales that are fundamental for comparative analyses. Traditional methods for large scale analysis often rely on simplified models that may overlook important spatial variability and uncertainty in flood hazard and impact estimates such as a simple aggregation of local flood depth maps and damage assuming homogeneous return periods (Alfieri et al., 2016; Vorogushyn et al., 2018).*
   *Lines 29-30*

*2) … and multivariate, assuming that flood damage is influenced by several variables including flood depth, inundation duration, flow velocity, and resistance parameters (e.g., building type, construction material) (Gerl et al., 2016).* Line 57

In the methodology, the authors mention that "We adopt here a uniform level of protection by assuming the threshold value $Q_*$ in Eq. (1) equal to design return period of the flood protection works." I found the approach interesting, but this is also a level of uncertainty, and I believe it is even more significant than the variation in manning coefficient. From an action planning and efficient decision-making point of view, the uncertainties associated with protection actions are more interesting and useful for evaluation than manning coefficient, right? Since it is important for decision-makers to know whether the protection actions adopted are effective or not, including from a spatial point of view. Manning is not a parameter that can be varied by them. Do you have any suggestions on how this could be incorporated into future work?

*Yes, the assumption of a "fixed threshold" for considering flood protection adds a component of uncertainty to the model. In our opinion, this uncertainty is less significant than the manning uncertainty, as the former generally affects the lower part of the damage distribution and for more frequent events the damage is expected to be much lower. The impact of the $Q_*$ uncertainty could be evaluated in a future work by varying the fixed threshold and comparing it with the manning uncertainty. An interesting alternative could be the use of spatially variable threshold, as proposed by Scussolini et al. (2016) using the FLOPROS database, which contains information on flood protection at different spatial scales. However, as the database is at a global scale, it would be better to have more detailed information, especially for areas such as the Adriatic where we do not find information on defense structures. In future work we could collaborate to obtain an accurate national scale database on flood protection in Italy with different information on the flood defense structure (length, height of the levee, spatial position etc.) to evaluate the effect of the structure at a local scale.*

*The following sentence was added to the revised text:*

*"While such assumption may add an additional component of uncertainty, we surmise that it is less significant than the uncertainty on the Manning parameter as it mainly affects the lower part of the damage distribution, which has a lesser impact on risk."*

Lines 129-131

Line 137 - Why use the rational formula? It usually has good applicability for small basins, but in your case the basins are large.

*We emphasize that our approach is not attached to a particular model for the assessment of discharge at the nodes, and we employ here the rational model for the sake of simplicity. The rational formula is a simplified but effective approach as it requires few input parameters (loss coefficient, rainfall intensity for a given return period scenario and time of concentration and basin area at the control section) that can be easily found or calibrated. Despite its application in generally small/medium sized basins, it is often applied to larger basins.*

*This concept is added to the paper as a continuation of the explanation of the hydrological load as follows:*

*"This approach is used because it is simple and flexible, and although it is usually applied to small catchments, it can be applied to larger catchments if it is properly calibrated". ==Lines 147-148==*

How were the C values determined for the different areas?

*The spatial variability of net rainfall (C\*rainfall height) was assessed using the modified Soil Conservation Service-Curve Number (SCS-CN) method of Hawkins et al. (2010). For each sub catchment (i.e. the subarea drained by each segment of the oriented network graph, i.e. the reference spatial unit for the hydrological modelling component), different values of average net precipitation are calculated. The CN map is the same calibrated and shown in Figure 7 in Pavesi et al., (2022). The catchments of the Adriatic area are mostly small and during the calibration we found that the use of a single C coefficient gave results more in line with the various official studies available online. For this reason, we preferred to change the methodology in this area. Thus, for about 75 % of the simulated territory, the modified SCS method was adopted, and for the remaining 25 %, a constant C coefficient was adopted.*

*This concept was already written in the paper but not comprehensible to the reader before reaching the "Model setup" section 3.1 so a reference to that section was added in ==Lines 146-147==.*

Is the study area well delimited and coincident with the entire area of the basin? Is there no "external" flow entering the basin?

*The study area is characterized by several catchments. The analysis was carried out on the 25 main basins defined by the Italian National Geoportal. The simulations were carried out separately for each of these basins. The basin to be simulated is then defined a priori and the area where the simulation is carried out coincides with the area of the basin. In this sense, there is no "external" flow entering the basin.*

*This concept was added in the "Model Setup" section as follows:*

*"The simulations were carried out separately for each of these basins. The basin to be simulated is then defined a priori and the area in which the simulation is carried out coincides with the area of the basin." ==Lines 291-293==*

Figure 1 – Is the hydrography in blue?

*Yes, a legend will be added in the revised manuscript to clarify this. ==Figure 1==*

I suggest including a brief description of the study area (with size) in the introduction, to familiarize the reader with the type of floods and scale that will be addressed. I later resolved many of my doubts about the methodology when I read the section on the study area and had a better understanding of the spatial scale.

*Thank you for your suggestion. It has been included in the revised manuscript.* Lines 80-82

Line 308 – C3 above 10^4 euro/year, right?

*Yes, it is. Thanks for the remark, C3 was added in the revised manuscript.*

Line 328

Line 303 – The authors states: "Note that agricultural land use has significantly different maximum damage values compared to the other land use types, varying by about two orders of magnitude." But the reader cannot note this, since the authors don't show this result, they present the risk value already in classes in Figure 2. Or does this variation in two orders of magnitude refer to the curves coming from Huizinga?

*The Huizinga curves are relative and not empirical damage curves, so they vary between 0 and 1. They are then associated with the maximum damage value, which is different for each country. In general, however, the maximum damage value for agriculture in European countries is on average two to three orders lower than that associated with residential, commercial and industrial land uses.*

Line 323-325

Either way, the authors provide many discussions in the next paragraphs in terms of LULC and the differences in the calculated maps and the ones obtained from PGRAAC. Therefore, I recommend including a figure of LULC in the study case section, to help the reader understand the discussions and draw their own conclusions.

*Thank you for your suggestion. Following the style of Figure 2 this new Figure is proposed:*

[Figure]

*Figure 3. Land use of the Central Apennines District (Italy) area at risk; panel a) shows a zoom on the upper Tiber River near the city of Perugia, panel b) shows a zoom on the middle Tiber valley near Rome, panel c) shows a zoom on the entire Musone River in the Marche Region.*

*Figure 3 and shown in Lines 345-347*

What is the advantage of using this proposed methodology in an area that already has risk maps constructed with resolutions and greater data accuracy? What new information is provided by the authors' approach? Or were the area chosen precisely to allow a comparison with reliable results and then expand the use of the methodology to other areas that do not have this detail with greater

reliability? I missed a bit of this discussion, and I think the innovation points presented in the introduction do not address these issues.

*Thank you for this comment, this is the core of the paper. In the introduction (line 84-90 of the original manuscript) we show the two main novelties of the paper. The discussion intrinsically explains both the aspects:*

1) *The advantage of using this proposed methodology in an area has risk maps constructed with resolution and greater data accurancy is twofold:*

    *- the methodology returns the distribution of risk in each cell rather than a deterministic value of risk; the maps produced take into account the uncertainty of the parameters (in the paper only the uncertainty of the Manning parameter is considered, but it can be extended to several sources). The idea of being able to perform uncertainty analyses on one or more model components provides a non-deterministic estimate and a range of confidence in the results, providing a more informed result for stakeholders and decision makers;*

    *- the hydrological-hydraulic analysis is carried out on a large scale and provides a comprehensive and homogeneous, albeit simplified, view of the hydrological-hydraulic processes affecting a whole area or region, rather than a local detailed analysis, which does not always have less uncertainty because complex analyses require more data to be input.*

2) *The qualitative comparison with PGRAAC maps shows that this type of simplified approach can be expanded to other areas. A simplified methodology such as RESCUE-FR is easily applicable at much larger scales, such as national and transnational scales, where input parameters of detailed models are even more difficult and uncertain to obtain and computational times are very large that can achieve months. The results of this methodology are neither qualitative nor deterministic. The risk distribution outcomes are expressed in economic terms (€/year), which are highly beneficial for stakeholders and the insurance sector. This approach provides them with a reliable range for their analyses, as opposed to a deterministic value that overlooks the uncertainties associated with hydrological and hydraulic variables, and damage models.*

    *Thank you for your suggestion, this was added in* ==*Lines 425-430.*==

Line 362 – There is a repeated "the".

   *Thank you for your careful scrutiny of our manuscript.* ==*Line 384*==

Bibliography

Pavesi, L., D'Angelo, C., Volpi, E., & Fiori, A. (2022). RESCUE: A geomorphology-based, hydrologic-hydraulic model for large-scale inundation mapping. *Journal of Flood Risk Management*, *15*(4), e12841.

Scussolini, P., Aerts, J. C., Jongman, B., Bouwer, L. M., Winsemius, H. C., de Moel, H., & Ward, P. J. (2016). FLOPROS: an evolving global database of flood protection standards. *Natural Hazards and Earth System Sciences*, *16*(5), 1049-1061.

**Reviewer #2**

Pavesi et al have presented an estimation of expected regional flood risk taking into account uncertainty within the Manning coefficient. While the uncertainty is considered for only one parameter, the framework resented by the authors can be used to account for other uncertainties. The analysis provided is rigorous and comprehensive, and I only have a few minor comments to further improve the manuscript.

*Dear reviewer,*

*Thank you for your careful and insightful review of our manuscript. We greatly appreciate the time and effort you have taken to provide such detailed comments and suggestions.*

1. Is the same Manning coefficient used across the entire spatial domain in a single simulation?

*Yes, it is. To clarify this, point a sentence was added in the revised manuscript.*

==Lines 296-297==

2. If the same Manning coefficient is used across the entire spatial region, it will be interesting to see a plot of regional risk R_n with respect to Manning coefficient.

[Figure]

*In the figure, Rn generally increases with increasing manning, because in the hazard model (RESCUE) the higher the roughness, the higher the water levels. The vulnerability curves vary with water depth and therefore the damage increases with increasing manning. Consequently, Rn also increases. The fact that the manning value of $5*10^{-3}$ ($s/m^{1/3}$) has a higher Rn than the immediately following values is due to the way the supercritical flow condition is treated in the RESCUE model.*

*How Rn varies with the Manning could be added in the Results section in the following way:*

*"In general Rn increases with increasing Manning's roughness because greater roughness leads to higher water depth. The water depth increases monotonically with the damage in the damage model adopted."* ==Lines 385-387==

3. Line 120 - DTM is defined after its first usage.

*Thank you, it is definite in line 130 so it needs to be changed.* ==Line 124==

4. Line 155 - It would be helpful for the reader if the authors elaborated on justification for their assumption - Uncertainty in Manning coefficient is considered as representative of most of the uncertainty characterizing the hydraulic modeling of flood risk.

*As explained in Annis et al. (2020) floodplain roughness is the main factor of uncertainty impacting the hydraulic modeling. This is represented in RESCUE-FR by the Manning coefficient. The uncertainty in the Manning coefficient is considered to be representative of most of the uncertainty that characterizes the hydraulic modelling of flood risk, because within the Manning uncertainty evaluated in the RESCUE-FR framework, different components of uncertainty are taken into account, such as the uncertainty in the DTM elevation, which affects the bottom channel slope and the extent of flood prone areas. Indeed, as explained in Johnson et al. (2019), low relief areas in a HAND-based model affect the results in terms of flood extents. This is because the HAND layers will be unable to accurately capture flooding extents due to essentially 0 relief. This can be taken into account considering lower Manning (left tail of the normal distribution of the Manning parameter). In addition, higher manning can also be interpreted as the probability that the reach will contract due to the possible presence of infrastructure that can change the water depth. In this sense, the uncertainty in the Manning coefficient is considered to be representative of most of the uncertainty that characterizes hydraulic modelling of flood risk.*

*This concept can be added to the paper as follows:*

*"Indeed, floodplain roughness is the main factor of uncertainty impacting the hydraulic modeling (Annis et al. 2020). This is represented in RESCUE-FR by the Manning coefficient. It represents a significant proportion of the overall uncertainty in hydraulic flood risk modelling, as it incorporates factors such as variations in terrain elevation, the extent of flood prone areas and potential channel restrictions due to the presence of infrastructure."*

==Lines 164-167==

5. Line 178 - The choice of normal distribution for Manning coefficient appears arbitrary. The authors have only stated that uniform distribution used by Pavesi et al. (2022) resulted in overestimation of

water depth, hence they selected the normal distribution. The appropriate distribution should be chosen based on the inherent uncertainty of the parameter, instead of the distribution's impact on the result. Is there any literature that can be referenced for the appropriate probability distribution of Manning coefficient?

*We are not aware of reliable estimates of the appropriate pdf for Manning. As explained in Stephens et al. (2020) there is no theoretical calculation of friction parameters, the use of empirical formulae, visual inspection and expert judgement makes quantitative prescriptions of this parameter inherently subjective. In Papaioannou et al. (2017), there is a comparison between the distribution of local manning values obtained from empirical formulae and different distributions (normal, lognormal, gamma, beta, etc.) on a local area in Greece. In terms of different goodness of fit statistics, different distributions can be chosen and the results between lognormal, gamma, normal and beta distributions are not so different, so the chosen distribution is basically a modeller's choice. In a large area it is difficult to obtain an empirical Manning distribution and to assess the inherent uncertainty of the parameter fitting different distributions. For this and the other reasons described in the paper, a symmetric normal distribution was chosen.*

*To clarify this concept, we could add the following statement to the paper:*

*"In general, there is no widely accepted probability distribution for the Manning coefficient, as its estimation relies heavily on empirical formulae and expert judgement, making it inherently subjective (Stephens et al., 2020). Comparative studies such as Papaioannou et al. (2017) show that normal, lognormal, gamma and beta distributions can all be used with minimal difference in results, leaving the choice of distribution to the expert judgement of the modeller."* ==Lines 189-193==

6. Line 182 - Parenthesis around DTM are unnecessary.

*Corrected.* ==Line 198==

7. Line 205 - between all the country in which -> among all the countries for which

*Corrected.* ==Line 221==

8. Line 207 - Is there supposed to be `and` between GDP and PPS?

*In Huizinga, (2007) it is defined in this way 'GDP per capita PPS'.*

9. Line 308 - classed -> classes.

*Thank you.* ==Line 329==

10. Line 285 - What is the purpose of this statement? - Where Q100 is the 1 − 1/100 = 0.99 quantile of the probability distribution of the peak discharge.

*It is to remark how Q100 is defined. To clarify this point, we added a parenthesis statement in* ==Line 305==.

*Thank you for your careful scrutiny of our manuscript.*

Bibliography

Annis, A., Nardi, F., Volpi, E., & Fiori, A. (2020). Quantifying the relative impact of hydrological and hydraulic modelling parameterizations on uncertainty of inundation maps. Hydrological Sciences Journal, 65(4), 507-523.

Johnson, J. M., Munasinghe, D., Eyelade, D., and Cohen, S.: An integrated evaluation of the national water model (NWM)–Height above nearest drainage (HAND) flood mapping methodology, Natural Hazards and Earth System Sciences, 19, 2405–2420, 2019.

Papaioannou, G., Vasiliades, L., Loukas, A., & Aronica, G. T. (2017). Probabilistic flood inundation mapping at ungauged streams due to roughness coefficient uncertainty in hydraulic modelling. *Advances in Geosciences*, *44*, 23-34.

Stephens, T. A., & Bledsoe, B. P. (2020). Probabilistic mapping of flood hazards: Depicting uncertainty in streamflow, land use, and geomorphic adjustment. *Anthropocene*, *29*, 100231.